# Quantifying tobacco and alcohol imagery in Netflix and Amazon Prime instant video original programming accessed from the UK: a content analysis

Alexander B Barker, Jordan Smith, Abby Hunter, John Britton, Rachael L Murray

UK Centre for Tobacco and Alcohol Studies, Division of Epidemiology and Public Health, University of Nottingham, Nottingham, UK

**Correspondence to**
Dr Alexander B Barker;
alexander.barker@nottingham.ac.uk

## ABSTRACT

**Objectives** Exposure to tobacco and alcohol content in audio-visual media is a risk factor for smoking and alcohol use in young people. Previous UK research has quantified tobacco and alcohol content in films and broadcast television but not that of video-on-demand (VOD) services such as Netflix and Amazon Prime. Furthermore, it is not clear whether regulation by Dutch (Netflix) or UK (Amazon Prime) authorities results in differences in content. We report an analysis of tobacco and alcohol content in a sample of episodes from the most popular programmes from these two VOD providers, and compare findings with earlier studies of UK prime-time television content.

**Setting** UK.

**Participants** None. Content analysis of a sample of 50 episodes from the five highest rated series released on Netflix and Amazon Prime in 2016, using 1 min interval coding of any tobacco or alcohol content, actual or implied use, paraphernalia and branding.

**Results** Of 2704 intervals coded, any tobacco content appeared in 353 (13%) from 37 (74%) episodes. Any alcohol content appeared in 363 (13%) intervals in 47 (94%) episodes. There were no significant differences between the two services, however the proportion of episodes containing tobacco and alcohol was significantly higher in VOD original programmes than those recorded in an earlier study of prime-time UK television.

**Conclusions** Audio-visual tobacco and alcohol content is common in VOD original programmes and represents a further source of exposure to imagery causing smoking uptake and alcohol use in young people. This appears to be equally true of services regulated in the UK and The Netherlands. Given that VOD services are consumed by a global audience, it appears likely that VOD content is an important global driver of tobacco and alcohol consumption.

## BACKGROUND

Preventing smoking uptake and alcohol use in young people is a public health priority, and there is now strong evidence that exposure to tobacco or alcohol advertising or other audio-visual content in the media increases uptake and subsequent use in adolescents.[1–11] However, while previous studies have

---

**Strengths and Limitations of this study**

► This study is the first to explore alcohol and tobacco content in video-on-demand (VOD) programmes.
► Established methods were used to explore the content in VOD original content.
► This study provides a comparison of VOD alcohol and tobacco content to UK broadcast television content.
► This study is limited to a sample of programmes and episodes on each VOD service.
► As viewing figures are not available for VOD original content, we could not estimate exposure to tobacco and alcohol content.

---

quantified tobacco and alcohol imagery in films and broadcast television, viewing habits are changing and online video-on-demand (VOD) services such as Netflix and Amazon Prime Instant Video, which allow users to watch whatever they choose at any time of day, are becoming more popular.[12] In the UK, an estimated 6.1 million households have access to Netflix[13] and 8 million to Amazon Prime Instant Video,[14] while Netflix now has 125 million paying subscribers worldwide.[15] In the UK, a fifth of people aged 16–24 years viewing time is spent watching VOD services and 46% of teenagers use Netflix, as opposed to 31% of adults.[16] Young people aged 5–16 years are now more likely to have watched a programme on VOD than on conventional TV channels such as BBC1 or ITV.[17]

Since 2015 VOD services based in the UK, including Amazon Prime Instant Video, have been subject to Office of Communications (Ofcom) regulations similar to those covering terrestrial television channels. These stipulate that 'smoking and misuse of alcohol must not be condoned, encouraged or glamorised in programmes likely to be widely seen by under-18s unless there is editorial justification'[18] (Section 1.10). Services available but not based in the UK are however

BMJ

not regulated by Ofcom but by regulations of the country within which they are registered.[19][20] Netflix, which is based in Amsterdam,[21] is subject to rules set out by the European Regulators Group for Audio-visual Media Regulators, including the European Audio-visual Media Services Directive,[22] which applies controls on content deemed harmful to young people but does not comment on non-commercial portrayals of tobacco or alcohol use in programme content.[23] It has previously been reported that Netflix programming includes more tobacco content than regular broadcast television in the USA,[24] but there are to date no studies of alcohol and tobacco content in VOD in the UK. We have therefore quantified tobacco and alcohol imagery in a sample of episodes from the most popular original programme series shown on two online VOD platforms, Amazon Prime Instant Video and Netflix, and compared our findings with those of earlier analyses of UK terrestrial television content.[25][26]

## METHODS

Since viewing figures are not made available for shows on Netflix and Amazon Prime Instant Video, ratings of their original shows from the Internet Movie Database,[27] IMDb (data correct as of March 2018), were used to identify the top five programme series from each service for the year 2016. We then took a systematic sample of five episodes from each series (the first, last and three episodes equally distributed through the series). To measure tobacco and alcohol content we used 1 min interval coding, a semiquantitative method used extensively in previous studies,[28–30] coding each interval for the presence of alcohol and tobacco content in the following categories:

### Actual use
Use of tobacco or alcohol on screen by any character.

### Implied use
Any inferred tobacco or alcohol use without any actual use on screen.

### Tobacco paraphernalia/other alcohol reference
The presence on screen of tobacco or alcohol or related materials.

### Brand appearance
The presence of clear and unambiguous tobacco or alcohol branding.

Tobacco and alcohol content were recorded as present in the 1 min interval if there was one appearance of any category in that interval. Multiple instances of the same category in the same interval were recorded as one event, but if the same event overlapped two intervals, this was coded as two separate events. 10 episodes (20%) were coded separately by two authors to ensure accuracy and reliability in the coding method. Data coding was completed in Microsoft Excel and, on completion, data were entered into IBM SPSS Statistics 24 for statistical analysis.

To investigate any potential differences regulation of the online services might have on tobacco and alcohol imagery, we used t-tests to compare mean levels of alcohol and tobacco content on each service. We also compared content with findings of our earlier studies of tobacco and alcohol content in prime-time UK television broadcast in 2015 using a $\chi^2$ analysis.

### Patient and public involvement
No patients were involved in this study.

## RESULTS

Details of the five highest rated programmes from each service are listed in online supplementary table 1 . A total of 50 episodes, comprising a total of 2704 1 min intervals (1325 and 1379 on Netflix and Amazon Prime Instant Video, respectively) were coded. The most common genre coded for was 'Drama' (six original programmes), followed by 'Action', 'Biography', 'Crime' and 'Comedy' which each had one original programme each, according to genre information taken from the Internet Movie Database.[27] Eight out of the 10 series were classified as '15' and two were classified as '12', according to the British Board of Film Classification.[31] Information about the series and episodes explored as part of the content analysis can be found in online supplementary table 1. The average amount of 1 min intervals per episode was 54.

There were no statistically significant differences between Amazon Prime Instant Video and Netflix original programmes in terms of the number of episodes or intervals containing either tobacco or alcohol content.

### Tobacco
Tobacco content occurred in 353 intervals (13% of all intervals) across 37 episodes (74% of all episodes) (figure 1). The average number of intervals containing tobacco content per episode was seven. In total, 97 intervals containing tobacco content also contained alcohol content (27% of all tobacco content intervals).

The most common category of tobacco content observed was actual tobacco use, which appeared in 246 intervals (9% of all intervals) across 33 episodes (66% of

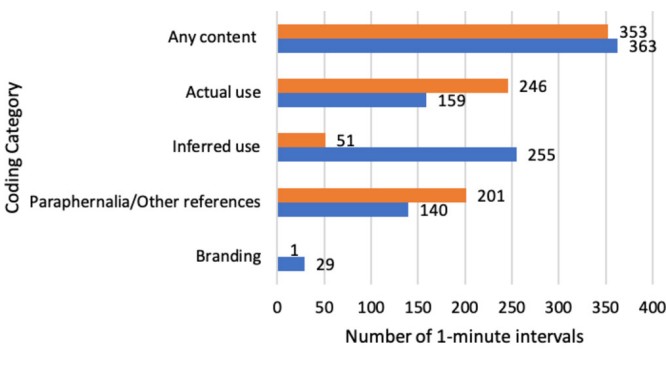

**Figure 1** Number of 1 min intervals containing tobacco and alcohol content by coding category.

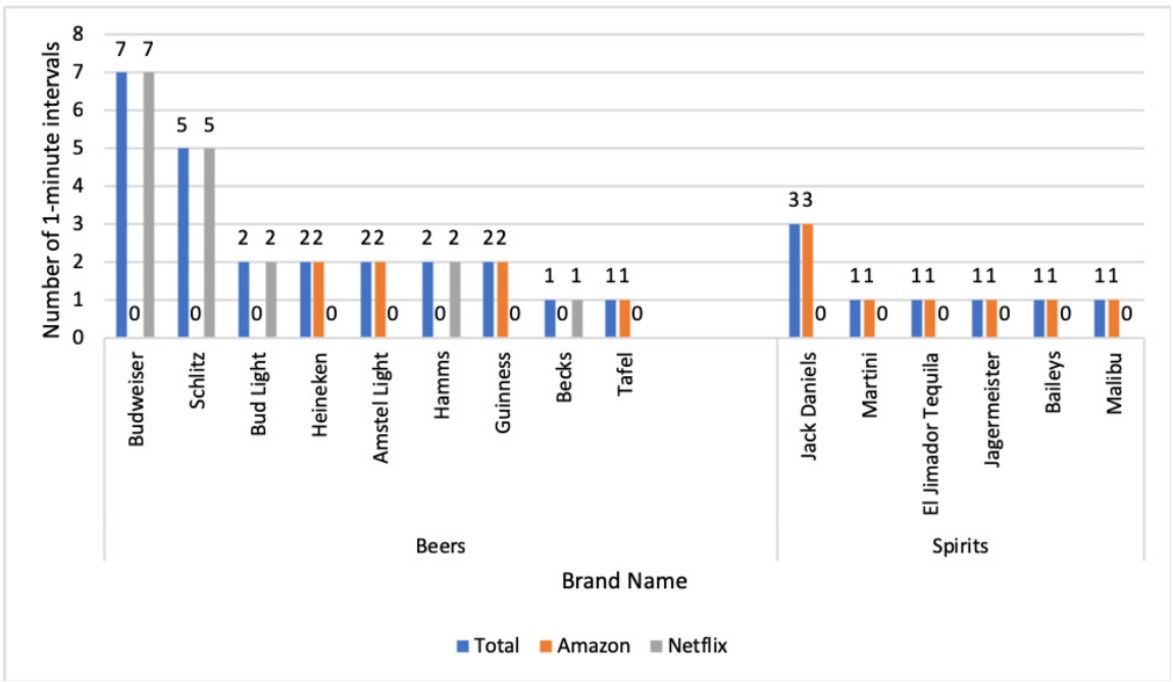

**Figure 2** Alcohol branding seen in Netflix and Amazon prime instant video original programming.

all episodes). The most prevalent form of actual tobacco use was cigarette smoking, which appeared in 228 intervals (93% of intervals containing tobacco use). Three intervals showed tobacco use by characters who were under-18. Netflix original programmes showed actual tobacco use in 144 1 min intervals (59% of intervals from Netflix), and Amazon Prime Instant Video in 102 1 min intervals (41% of intervals from Amazon Prime Instant Video). The programme which showed the most actual tobacco use was 'The Crown' on Netflix, with 60 intervals (24% of all intervals from 'The Crown').

Inferred tobacco use occurred in 51 intervals (2% of all intervals), across 17 episodes (34% of all episodes). The most prevalent inferred tobacco use was non-verbal, such as characters holding but not visibly smoking cigarettes, which appeared in 45 intervals (88% of inferred tobacco use intervals). Tobacco paraphernalia appeared in 201 intervals (7% of all intervals) across 34 episodes (68% of all episodes), most commonly ashtrays, which were seen in 133 intervals (66% of intervals containing paraphernalia).

Tobacco branding, involving the 'Camel' branded cigarette pack occurred once in an episode of 'Stranger Things' on Netflix.

### ALCOHOL

Alcohol content appeared in 363 intervals (13% of all intervals) across 47 episodes (94% of all episodes). The most prevalent type of alcohol content was inferred use (figure 1). The average number of intervals containing alcohol content per episode was seven. In total, 97 intervals containing alcohol content also contained tobacco content (27% of all tobacco alcohol intervals).

Actual alcohol use appeared in 159 intervals (6% of all intervals) across 40 episodes (80% of all episodes). The type of alcohol most commonly consumed was spirits (65 of all intervals, 41% of actual use intervals). Two intervals contained alcohol use by characters who were under 18. Of all 1 min intervals containing actual alcohol consumption, 86 (54%) were in Amazon Prime Instant Video original programmes and 73 (46%) were in Netflix original programmes. The programme with the most 1 min intervals containing actual alcohol use was Goliath, on Amazon Prime Instant Video, with 29 1 min intervals (18% of all intervals containing actual alcohol use).

Inferred alcohol use was seen in 255 intervals (9% of all intervals) across 43 episodes (86% of all episodes), most commonly in the form of characters holding alcohol drinks (221 intervals, 87% of intervals containing inferred alcohol use). Other alcohol references appeared in 140 intervals (5% of all intervals) across 35 episodes (70% of all episodes), most commonly alcohol bottles being shown on screen (111 intervals, 79% of intervals containing other alcohol references).

Branding was seen in 29 intervals (1% of all intervals) in 16 episodes (32% of all episodes), across nine programmes. Sixteen different brands were seen, the most prevalent was 'Budweiser' (seen in seven intervals), however, this brand was only seen in original programming from Netflix. All occurrences of 'spirit' branding was seen on Amazon Prime Instant Video, whereas the majority of 'beer' branding was seen on Netflix. No wine brands were observed. (figure 2). One of the most prominent brands, Schlitz, appeared in only one programme series, Stranger Things.

**Table 1** Proportion (%) of programmes containing tobacco or alcohol imagery in VOD and UK broadcast television

|  | Video-on-demand | Television[25 26] |
|---|---|---|
| **Tobacco** | | |
| Any tobacco content | 74% | 17%* |
| Actual tobacco use | 66% | 5%* |
| Implied tobacco use | 34% | 6.8%* |
| Tobacco paraphernalia | 68% | 12%* |
| Tobacco branding | 2% | 0.6% |
| **Alcohol** | | |
| Any alcohol content | 94% | 54%* |
| Actual alcohol use | 80% | 11%* |
| Implied alcohol use | 70% | 38%* |
| Other alcohol reference | 86% | 40%* |
| Alcohol branding | 32% | 13%* |

*Differences were statistically significant (p≤0.05).

### Comparison between VOD services and terrestrial television content

Comparison with earlier content analyses of UK broadcast television[25 26] however shows that the proportion of episodes containing tobacco and alcohol was significantly higher in VOD services (table 1).

### DISCUSSION

This study demonstrates that tobacco and alcohol content is similarly common in original programming on VOD services regulated by different national authorities, and that in comparison with studies of prime-time UK television includes significantly more tobacco and alcohol imagery. Since previous research evaluating the effect of alcohol and tobacco in the media on initiation of alcohol and tobacco use shows a dose-response relation,[32–34] our findings indicate that in addition to conventional terrestrial television, original programming on VOD services is a potentially important source of exposure to tobacco and alcohol imagery. The findings also suggest that the apparently stricter regulations applied by Ofcom in the UK relative to those applied to Netflix in the Netherlands do not translate into appreciably lower levels of content.

We found examples of alcohol brands regularly occurring in programmes from each VOD service. Amazon Prime instant Video is regulated by Ofcom and is subject to the Ofcom Broadcasting Code.[35] Section 9.13 of the code specifies that the product placement of alcoholic drinks is prohibited. However, we concede that if the programme makers acquired these products at no significant costs, they could be considered as props according to the code.[35]

This study is the first content analysis of tobacco and alcohol imagery in original programming on VOD services. This study used semiquantitative 1 min interval coding methods, established in previous studies on tobacco and alcohol content in the media,[30 36–45] allowing these results to be compared with others. For logistic reasons we were able to code only a sample of episodes from a sample of original programmes. Ideally, we would have chosen the programmes with the highest levels of viewing but in the absence of VOD viewing figures chose the series rated most highly on an independent online database on the assumption that these were likely also to be the most widely viewed. We accept that our sample is small, and therefore may not be fully representative of content offered by these VOD providers, and that in at least one programme series (The Crown) the high levels of smoking portrayed may have been historically representative. We also recognise that in the absence of detailed viewing figures, we do not know the extent to which the series sampled were popular with younger viewers.

We have previously quantified tobacco and alcohol content in prime-time UK broadcast television from 2015, and demonstrated that tobacco content appeared in 3% and alcohol in 12% of intervals.[30 43] While there are differences in the sampling of programmes, with the previous studies coding all broadcasts on terrestrial television during specified time periods, it is concerning that the proportion of episodes containing tobacco and alcohol in VOD content was appreciably higher,[25] indicating that original programming on VOD services is an important yet overlooked source of exposure to tobacco and alcohol imagery.

Our findings therefore indicate that VOD services are likely to be a further source of harmful exposure to tobacco and alcohol imagery in children. Given the ease of access to programming for children and adolescents, regulators need to account for the changing viewing habits of adolescents in order to protect this group from potentially harmful imagery via VOD services.

Our study is limited to VOD programmes that are popular in the UK, but these services reach global audiences and it is likely that our findings apply much more widely than in the UK. In the UK there is scope under the regulatory powers exercised by Ofcom since 2015[46] to reduce this content, at least for UK-based VOD companies. However, VOD services are consumed by a global audience, and given the ease of access to programming for children and adolescents, UK and other national regulators need to account for the changing viewing habits of adolescents to protect this group from potentially harmful imagery via VOD services.

While it is likely VOD services are a source of harmful exposure to tobacco and alcohol imagery in children, establishing the extent to which this is true requires data on viewing of these services by children, and content data on the programmes that are most widely seen by young people. Future research should also explore potentially harmful content in a larger sample of programmes.

**Contributors** AB and RM designed the study. JS, AH and AB collected data for the study. AB, RM and JB wrote the manuscript.

**Funding**  This work was supported by Medical Research Council (grant number MR/K023195/1) for the UK Centre for Tobacco and Alcohol Studies, which includes funding from the British Heart Foundation, Cancer Research UK, Economic and Social Research Council and the Department of Health under the auspices of the UK Clinical Research Collaboration. The funders had no role in the study design, data collection and analysis, decision to publish or preparation of the manuscript.

**Competing interests**  None declared.

**Patient consent for publication**  Not required.

**Provenance and peer review**  Not commissioned; externally peer reviewed.

**Data sharing statement**  Additional data on the tobacco and alcohol content found in each episode of each programme are available on request from Alexander. barker@nottingham.ac.uk.

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
