## [Reviewer comments · BMJ Open]

ARTICLE DETAILS

TITLE (PROVISIONAL)	Quantifying tobacco and alcohol imagery in Netflix and Amazon Prime Instant Video original programming accessed from the UK: a content analysis
AUTHORS	Barker, Alexander; Smith, Jordan; Hunter, Abby; Britton, John; Murray, Rachael

VERSION 1 – REVIEW

REVIEWER	Rob McGee Department of Preventive and Social Medicine, Dunedin School of Medicine, University of Otago, Dunedin, New Zealand
REVIEW RETURNED	20-Sep-2018

GENERAL COMMENTS	Thank you for the opportunity to review this paper. I believe it is important to document the extent to which small-screen viewers are exposed to tobacco and alcohol imagery in the programmes they view. Overall, I think the Introduction is well reasoned and clearly states why this kind of research is important in the UK context. One suggestion is to omit the acronym AVC and just refer to "imagery"; AVC is only used a handful of times in the text. The methods look to be straightforward and appropriate for the study design. My only concern would be with the reliability of the coding. Were there any checks on rater consistency (if one person did all the ratings) or inter-rater reliability (if there were multiple raters)? This might be especially important in cases of implied use. With respect to the Results, I would make the following points. First, I would suggest that Table 1 could be summarised in the text. Unless the reader intends to follow up specific episodes, describing those coded seems unnecessary. As the age ratings are nearly all 15+, this could be reported in text. Not sure about providing the programme names but again doesn't seem essential in interpreting the findings. Second, Tobacco content is reported as present in 353 intervals of the 2704 total; that's 13% (not 12%); this is also in the Abstract. Third, at the outset of the results, the authors could report that there were no statistically significant differences between the two VOD providers in tobacco or alcohol content. This being so, it may be better to combine the two providers data for any reported figures. If this was done, Figures 1 and 2 could be combined to contrast the amount of alcohol and tobacco imagery in the same Figure. It is of note that there seems to be the same amount of tobacco and alcohol imagery, and if anything a bit more tobacco imagery. The situation is totally the opposite in NZ where we have
--

	done some similar work with NZ free to air tv; alcohol imagery far outweighs tobacco imagery. In Figure 3 it might be more useful to combine the beer data and contrast with the spirit data; it looks like beer branding is shown more often than spirits. Was no wine branding visible? Is it the case that Figure 3 represents a kind of "product placement"? Finally in Table 2, the p-value column could be safely omitted and a footnote added to the table indicating which differences were significant. Its important as noted in the discussion that the VOD imagery far outweighs the UK tv imagery. And especially in brand content which seems to amount to unpaid advertising. Do the authors have information on the extent to which alcohol and tobacco imagery occurred concurrently in the same interval? Overall, I think this paper provides interesting findings, but some content could be summarised or omitted to emphasise the important findings.
--	---

REVIEWER	Kate Hunt University of Stirling
REVIEW RETURNED	05-Oct-2018

GENERAL COMMENTS	Comments for authors: I enjoyed reading this paper which is well-written and makes an important contribution to the literature on (young) people's exposure to images of tobacco and alcohol. I only have minor suggestions for improvement, Backgrounds: Hanewinkel et al have also published on exposures to Tobacco as well as alcohol, and on the uptake of tobacco one year following reported exposures to movie images in young people. . These findings are also relevant to cite here. I think it would be helpful in include number of intervals per episode/ series I appreciate that it would still be an underestimate of exposures but presumably there were multiple intervals within episode which included instances of smoking/drinking, but I think reporting the mean number of intervals which include smoking/drinking would add to the paper. In the results section, please could the authors check carefully that it is always clear what % are of. It is not always clear whether they are within episodes, series, or the total sample. On p7, the authors state that "two intervals contained alcohol use in under-18s" but my understanding from table 1 was that all series are for under-18s all but One are rated 15 and the last is a 12.
---

REVIEWER	Kate Hunt University of Stirling
REVIEW RETURNED	05-Oct-2018

GENERAL COMMENTS	Comments for authors: I enjoyed reading this paper which is well-written and makes an important contribution to the literature on (young) people's
---

	exposure to images of tobacco and alcohol. I only have minor suggestions for improvement, Backgrounds: Hanewinkel et al have also published on exposures to Tobacco as well as alcohol, and on the uptake of tobacco one year following reported exposures to movie images in young people. . These findings are also relevant to cite here. I think it would be helpful to include number of intervals per episode/ series I appreciate that it would still be an underestimate of exposures but presumably there were multiple intervals within episode which included instances of smoking/drinking, but I think reporting the mean number of intervals which include smoking/drinking would add to the paper. In the results section, please could the authors check carefully that it is always clear what % are of. It is not always clear whether they are within episodes, series, or the total sample. On p7, the authors state that "two intervals contained alcohol use in under-18s" but my understanding from table 1 was that all series are for under-18s all but One are rated 15 and the last is a 12.
--	--

VERSION 1 – AUTHOR RESPONSE

Reviewer(s)' Comments to Author:

Reviewer: 1

Reviewer Name: Rob McGee

Institution and Country: Department of Preventive and Social Medicine, Dunedin School of Medicine, University of Otago, New Zealand

Please state any competing interests or state 'None declared': none declared

Please leave your comments for the authors below

Thank you for the opportunity to review this paper. I believe it is important to document the extent to which small-screen viewers are exposed to tobacco and alcohol imagery in the programmes they view.

Overall, I think the Introduction is well reasoned and clearly states why this kind of research is important in the UK context. One suggestion is to omit the acronym AVC and just refer to "imagery"; AVC is only used a handful of times in the text.

We have now replaced this acronym with the word imagery as suggested.

The methods look to be straightforward and appropriate for the study design. My only concern would be with the reliability of the coding. Were there any checks on rater consistency (if one person did all the ratings) or inter-rater reliability (if there were multiple raters)? This might be especially important in cases of implied use.

We apologise for causing concern, the following sentence has been added to the methods section.

'10 episodes (20%) were coded separately by two authors to ensure accuracy and reliability in the coding method.' – Page 5, Lines 17-18.

With respect to the Results, I would make the following points. First, I would suggest that Table 1 could be summarised in the text. Unless the reader intends to follow up specific episodes, describing those coded seems unnecessary. As the age ratings are nearly all 15+, this could be reported in text. Not sure about providing the programme names but again doesn't seem essential in interpreting the findings.

I agree that the information contained in table 1 is not essential for interpreting the results, however, I do feel that this information is important for the reader to understand which series and episodes were explored as part of the content analysis. I have further described the information from Table 1 in the text;

'Details of the five highest rated programmes from each service are listed in Supplementary Table 1. A total of 50 episodes, comprising a total of 2704 one-minute intervals (1325 and 1379 on Netflix and Amazon Prime Instant Video, respectively) were coded. The most common genre coded for was 'Drama' (6 original programmes), followed by 'Action', 'Biography', 'Crime', and 'Comedy' which each had 1 original programme each, according to genre information taken from the Internet Movie Database. Eight out of the ten series were classified as '15' and two were classified as '12', according to the British Board of Film Classification. Information about the series and episodes explored as part of the content analysis can be found in supplementary file 1.' – Page 6, lines 2-10.

I have changed table 1 from being present in the text to a supplementary file, so that the reader has access to this information if it is required.

Second, Tobacco content is reported as present in 353 intervals of the 2704 total; that's 13% (not 12%); this is also in the Abstract.

We apologise for this error, this has now been corrected in the abstract and the main text.

Third, at the outset of the results, the authors could report that there were no statistically significant differences between the two VOD providers in tobacco or alcohol content. This being so, it may be better to combine the two providers data for any reported figures. If this was done, Figures 1 and 2 could be combined to contrast the amount of alcohol and tobacco imagery in the same Figure. It is of note that there seems to be the same amount of tobacco and alcohol imagery, and if anything a bit more tobacco imagery. The situation is totally the opposite in NZ where we have done some similar work with NZ free to air tv; alcohol imagery far outweighs tobacco imagery.

We have now moved the following paragraph to the start of the results section

'There were no statistically significant differences between Amazon Prime Instant Video and Netflix original programmes in terms of the number of episodes or intervals containing either tobacco or alcohol content.' – page 6, lines 12-14.

We have now combined data on the amount of tobacco and alcohol in the same figure. References to figure 2 in the text have now been changed to figure 1 accordingly.

In Figure 3 it might be more useful to combine the beer data and contrast with the spirit data; it looks like beer branding is shown more often than spirits. Was no wine branding visible? Is it the case that Figure 3 represents a kind of "product placement"?

We have edited figure 3 (now figure 2) so that the different brands are categorised between beers and spirits. We note that all spirit branding was seen on Amazon Prime, whereas the majority of beer branding was seen on Netflix. We saw no wine brands. We agree that the inclusion of these brands in the programmes is suggestive of product placement.

We have added the following to the text;

'All occurrences of 'spirit' branding was seen on Amazon Prime Instant Video, whereas the majority of 'beer' branding was seen on Netflix. No wine brands were observed (Figure 3).' – page 7, lines 27-29.

We have also included the following discussion of this apparent product placement in the discussion section.

'We found examples of alcohol brands regularly occurring in programmes from each VOD service. Amazon Prime instant Video is regulated by Ofcom and is subject to the Ofcom Broadcasting Code. (35) Section 9.13 of the code specifies that the product placement of alcoholic drinks is prohibited. However, we concede that if the programme makers acquired these products at no significant costs, they could be considered as props according to the code.(35)' – page 9, lines 3-7.

Finally in Table 2, the p-value column could be safely omitted and a footnote added to the table indicating which differences were significant.

We have now omitted the P value column and added a footnote as suggested

Table 2: Proportion (%) of programmes containing tobacco or alcohol AVC in VOD and UK broadcast television.

Video-on-demand
Television (22, 23)

Tobacco

Any Tobacco Content	74%	17%*
Actual Tobacco Use	66%	5%*
Implied Tobacco Use	34%	6.8*
Tobacco paraphernalia	68%	12%*
Tobacco Branding	2%	0.6%

Alcohol

Any Alcohol Content	94%	54%*
Actual Alcohol Use	80%	11%*
Implied Alcohol Use	70%	38%*
Other Alcohol reference	86%	40%*
Alcohol Branding	32%	13%*

* Differences were statistically significant ($p < 0.05$)

Its important as noted in the discussion that the VOD imagery far outweighs the UK tv imagery. And especially in brand content which seems to amount to unpaid advertising. Do the authors have information on the extent to which alcohol and tobacco imagery occurred concurrently in the same interval?

We have revisited the data and have now provided information on the number of intervals containing both tobacco and alcohol content.

'In total, 97 intervals containing tobacco content also contained alcohol content (27% of all tobacco content intervals)' – page 6, lines 21-22.

'In total, 97 intervals containing alcohol content also contained tobacco content (27% of all tobacco alcohol intervals)' – page 7, lines 11-12.

Overall, I think this paper provides interesting findings, but some content could be summarised or omitted to emphasise the important findings.

We hope that the changes made helps to emphasise the important findings of the paper.

Reviewer: 2

Reviewer Name: Kate Hunt

Institution and Country: University of Stirling

Please state any competing interests or state 'None declared': None Declared

Please leave your comments for the authors below

Comments for authors:

I enjoyed reading this paper which is well-written and makes an important contribution to the literature on (young) people's exposure to images of tobacco and alcohol. I only have minor suggestions for improvement,

Backgrounds: Hanewinkel et al have also published on exposures to Tobacco as well as alcohol, and on the uptake of tobacco one year following reported exposures to movie images in young people. . These findings are also relevant to cite here.

We have extended the list of citations in regard to the effect of media content on uptake as suggested. The references now cited to support this claim are as follows;

1. Anderson P, De Brujin A, Angus K, Gordon R, Hastings G. Impact of alcohol advertising and media exposure on adolescent alcohol use: a systematic review of longitudinal studies. *Alcohol and Alcoholism*. 2009;44(3):229-43.
2. Smith L, Foxcroft DR. The effects of alcohol advertising, marketing and portrayal on drinkin behaviour in young people: systematic review of prospective cohort studies. *BMC Public Health*. 2009;9(51):1-11.
3. Hanewinkel R, Sargent JD, Hunt K, Sweeting H, Engels RC, Scholte RH. Portrayal of alcohol consumption in movies and drinking initiation in low-risk adolescents. *Pediatrics*. 2014;133:973-82.
4. Chang F, Miao N, Lee C, Chiu C, Lee S. The association of media exposure and media literacy with adolescent alcohol and tobacco use. *Journal of Health Psychology*. 2016;21(4):513-25.
5. Leonardi-Bee J, Nderi M, Britton J. Smoking in movies and smoking initiation in adolescents: systematic review and meta-analysis. *Addiction*. 2016;111(10):1750-63.
6. U.S. Department of Health and Human Services. Preventing Tobacco Use Among Youth and Young Adults: A Report of the Surgeon General. Atlanta, GA: U.S. Department of Health and Human Services, Centers for Disease Control and Prevention, National Center for Chronic Disease Prevention and Health Promotion, Office on Smoking and Health; 2012.
7. U.S. Department of Health and Human Services. The Health Consequences of Smoking: 50 Years of Progress: A Report of the Surgeon General. Atlanta, GA: U.S. Department of Health and

Human Services, Centers for Disease Control and Prevention, National Center for Chronic Disease Prevention and Health Promotion, Office on Smoking and Health; 2014.

8. National Cancer Institute. The Role of the Media in Promoting and Reducing Tobacco Use. Tobacco Control Monograph No. 19.: U.S. Department of Health and Human Services, National Institutes of Health; 2008.

9. Hanewinkel R, Sargent J. Exposure to smoking in popular contemporary movies and youth smoking in Germany. *American Journal of Preventative Medicine*. 2007;32(6):466-73.

10. Gendall P, Hoek J, Edwards R, Glantz S, Langevin SM. Effect of exposure to smoking in movies on young adult smoking in New Zealand. *PLoS One*. 2016;11(3):e0148692.

11. Morgenstern M, Poelan EA, Scholte R, Karlsdottir S, Jonsson SH, Mathis F, et al. Smoking in movies and adolescent smoking: cross-cultural study in six European countries. *Thorax*. 2011;66(10):875-83.

I think it would be helpful to include number of intervals per episode/ series. I appreciate that it would still be an underestimate of exposures but there were multiple intervals within episode which included instances of smoking/drinking, but I think reporting the mean number of intervals which include smoking/drinking would add to the paper.

We realise that this information could be useful and have now provided this information in the text;

'The average amount of 1-minute intervals per episode was 54.' – page 6, Lines 9-10

'The average number of intervals containing tobacco content per episode was 7.' – page 6, Line 20.

'The average number of intervals containing alcohol content per episode was 7.' – page 7, lines 10-11.

In the results section, please could the authors check carefully that it is always clear what % are of. It is not always clear whether they are within episodes, series, or the total sample.

We apologise for this confusion and have now clarified in the text what the percentages are in reference to;

'Tobacco

Tobacco content occurred in 353 intervals (13% of all intervals) across 37 episodes (74% of all episodes) (Figure 1).

Figure 1 here

The most common category of tobacco content observed was actual tobacco use, which appeared in 246 intervals (9% of all intervals) across 33 episodes (66% of all episodes). The most prevalent form of actual tobacco use was cigarette smoking, which appeared in 228 intervals (93% of intervals containing tobacco use). Three intervals showed tobacco use by characters who were under-18.

Netflix original programmes showed actual tobacco use in 144 one-minute intervals (59% of intervals from Netflix), and Amazon Prime Instant Video in 102 one-minute intervals (41% of intervals from Amazon Prime Instant Video). The programme which showed the most actual tobacco use was 'The Crown' on Netflix, with 60 intervals (24% of all intervals from 'The Crown').

Inferred tobacco use occurred in 51 intervals (2% of all intervals), across 17 episodes (34% of all episodes). The most prevalent inferred tobacco use was non-verbal, such as characters holding but not visibly smoking cigarettes, which appeared in 45 intervals (88% of inferred tobacco use intervals). Tobacco paraphernalia appeared in 201 intervals (7% of all intervals) across 34 episodes (68% of all episodes), most commonly ashtrays, which were seen in 133 intervals (66% of intervals containing paraphernalia).

Tobacco branding, involving the 'Camel' branded cigarette pack occurred once in an episode of 'Stranger Things' on Netflix.

Alcohol

Alcohol content appeared in 363 intervals (13% of all intervals) across 47 episodes (94% of all episodes). The most prevalent type of alcohol content was inferred use (Figure 2).

Figure 2 here

Actual alcohol use appeared in 159 intervals (6% of all intervals) across 40 episodes (80% of all episodes). The type of alcohol most commonly consumed was spirits (65 of all intervals, 41% of actual use intervals). Two intervals contained alcohol use by characters who were under-18. Of all one-minute intervals containing actual alcohol consumption, 86 (54%) were in Amazon Prime Instant Video original programmes and 73 (46%) were in Netflix original programmes. The programme with the most one-minute intervals containing actual alcohol use was Goliath, on Amazon Prime Instant Video, with 29 one-minute intervals (18% of all intervals containing actual alcohol use).

Inferred alcohol use was seen in 255 intervals (9% of all intervals) across 43 episodes (86% of all episodes), most commonly in the form of characters holding alcohol drinks (221 intervals, 87% of intervals containing inferred alcohol use). Other alcohol references appeared in 140 intervals (5% of all intervals) across 35 episodes (70% of all episodes), most commonly alcohol bottles being shown on screen (111 intervals, 79% of intervals containing other alcohol references).

Branding was seen in 29 intervals (1% of all intervals) in 16 episodes (32% of all episodes), across 9 programmes. Sixteen different brands were seen, the most prevalent was 'Budweiser' (seen in seven intervals), however, this brand was only seen in original programming from Netflix (Figure 3). One of the most prominent brands, Schlitz, appeared in only one programme series, Stranger Things.' – page 7, line 18 –line 31.

On p7, the authors state that "two intervals contained alcohol use in under-18s" but my understanding from table 1 was that all series are for under-18s all but One are rated 15 and the last is a 12.

We apologise for this confusion, we were not referring to the age rating of the programme but the perceived age of the character in the series consuming alcohol. We have edited this sentence to read;

'Two intervals contained alcohol use by characters who were under-18' – page 7, line 15 – page 8, line 1.

In light of this we have also edited the following sentence about tobacco use;

'Three intervals showed tobacco use by characters who were under-18.' – page 6, line 27.

VERSION 2 – REVIEW

REVIEWER	Rob McGee Department of Preventive and Social Medicine, University of Otago. Dunedin, New Zealand
REVIEW RETURNED	22-Nov-2018
GENERAL COMMENTS	I have carefully read the revised paper and I believe the authors have satisfactorily addressed the concerns I expressed in my first review. I was unable to access Figures 1 and 2 on the website, but these were satisfactorily presented in the first review. I recommended combining figures 1 and 2 and this seems to have been done.